# Pathogens and Antibiotic Susceptibilities of Global Bacterial Keratitis: A Meta-Analysis

**DOI:** 10.3390/antibiotics11020238

**Published:** 2022-02-12

**Authors:** Zijun Zhang, Kai Cao, Jiamin Liu, Zhenyu Wei, Xizhan Xu, Qingfeng Liang

**Affiliations:** Beijing Institute of Ophthalmology, Beijing Tongren Eye Center, Beijing Tongren Hospital, Capital Medical University, Beijing Key Laboratory of Ophthalmology and Visual Sciences, Beijing 100005, China; shenyu@ccmu.edu.cn (Z.Z.); caozhi@ccmu.edu.cn (K.C.); liujiamin@mail.ccmu.edu.cn (J.L.); weizhenyu@ccmu.edu.cn (Z.W.); xuxz0924@mail.ccmu.edu.cn (X.X.)

**Keywords:** bacteria, keratitis, microorganisms, antibiotic, susceptibility

## Abstract

Bacterial keratitis (BK) is the most common type of infectious keratitis. The spectrum of pathogenic bacteria and their susceptibility to antibiotics varied with the different regions. A meta-analysis was conducted to review the global culture rate, distribution, current trends, and drug susceptibility of isolates from BK over the past 20 years (2000–2020). Four databases were searched, and published date was limited between 2000 and 2020. Main key words were “bacterial keratitis”, “culture results” and “drug resistance”. Forty-two studies from twenty-one countries (35 cities) were included for meta-analysis. The overall positive culture rate was 47% (95%CI, 42–52%). Gram-positive cocci were the major type of bacteria (62%), followed by Gram-negative bacilli (30%), Gram-positive bacilli (5%), and Gram-negative cocci (5%). *Staphylococcus* spp. (41.4%), *Pseudomonas* spp. (17.0%), *Streptococcus* spp. (13.1%), *Corynebacterium* spp. (6.6%) and *Moraxella* spp. (4.1%) were the most common bacterial organism. The antibiotic resistance pattern analysis revealed that most Gram-positive cocci were susceptive to aminoglycoside (86%), followed by fluoroquinolone (81%) and cephalosporin (79%). Gram-negative bacilli were most sensitive to cephalosporin (96%) and fluoroquinolones (96%), followed by aminoglycoside (92%). In Gram-positive cocci, the susceptibility trends of fluoroquinolones were decreasing since 2010. Clinics should pay attention to the changing trends of pathogen distribution and their drug resistance pattern and should diagnose and choose sensitive antibiotics based on local data.

## 1. Introduction

Infectious keratitis (IK) is a potentially sight-threatening condition, which leads to at least 1.5 to 2 million new cases of unilateral blindness every year [1,2,3]. Among them, bacterial keratitis (BK) is the most common type according to the reports from multiple regions such as UK [4,5,6,7], North & South America [8,9,10,11], Middle East [12] and Australia [13,14]. Common risk factors for BK were contact lens wear, ocular trauma, ocular surface disease, and prior ocular surgery [15]. Bacterial culture via corneal scraping samples is still the gold standard for the diagnosis of BK, which permits isolation of the causal bacteria. However, not all medical institutions are able to carry out those tests due to various limitations. Therefore, it is critical for clinicians to make empirical diagnosis to know the pathogenic microorganism and antibiotics appropriate for eradicating the infection [16].

The common organisms that cause bacterial keratitis include *Staphylococcus aureus*, Coagulase-negative *staphylococci* (CoNS), *Streptococcus pneumoniae*, and others [17,18,19,20,21]. The bacterial spectrum and drug susceptibility for bacterial isolates from different areas or periods are widely reported [5,8,18,22,23], but the most common pathogen of BK remains debatable. *Pseudomonas* spp. were demonstrated to be the most common pathogen in Malaysia [22], Iran [23] and Taiwan [16], while CoNS are reported to be the most common in UK [5,6,24] and Australia [13]. Due to the widespread use of broad-spectrum antibiotics, it is very likely that the bacterial spectrum and its resistance to antibiotics varies greatly over time and from area to area. However, a comprehensive worldwide and long-term data analysis is scarce. To analyze the trends of bacteria and drug resistance profile over time in the world, we conducted a meta-analysis to compare the positive rate of culture in medical facilities worldwide and summarize the temporal and spatial trends of microbial isolates and their susceptibility patterns since 2000.

## 2. Results

### 2.1. Literature Search and Study Characteristics

From the selected databases, 4734 potentially relevant references were identified. In total, 1156 references were excluded because of duplicates. Details of searching and de-duplications were shown in Appendix B. Search results were shown in Appendix C by reviewing the titles and abstracts, and 3459 references were excluded. After reading 119 full texts, 16 articles were excluded, and 103 papers were assessed for eligibility. A total of 42 studies were ultimately included in this meta-analysis, of which only 38 articles contained enough information for positive rate analysis, and 33 articles for drug susceptibility analysis. We used the methodological scoring system of “rate” to assess the quality of each study. The score of all studies were more than 4 points. The paper selection process was shown in Figure 1, and the characteristics of the included studies were shown in Table 1. We used the methodological scoring system of “rate” to assess the quality of each study. The score of all studies were more than 4 points. The paper selection process was shown in Figure 1, and the characteristics of the included studies was shown in Table 1. Among these data, 16 articles were reported from Asia, 11 from America, 1 from Africa, 9 from Europe and 5 from Oceania.

### 2.2. Positive Rate of Culture

A total of 38,931 samples scraping from the cornea of BK patients were reviewed in the study. Among them, 14,596 samples were culture positive and positive rate of culture was 47% (95%CI, 42–52%) based on the 38 studies (Figure 2). The highest positive rate was 83% and the lowest positive rate was 21%. Data of positive rate was available among 21 countries (35 cities) in 5 continents. The highest and the lowest positive rate analyzed by counties were from Korea and from Turkey (83% vs. 28%, Figure 3, Appendix A). There were no significant differences of positive culture rate among different countries (*p* = 0.464). Grouped by continents, the highest positive rate was 59% (95%CI, 48–68%), found in 4 articles performed in Oceania and the lowest was 40% (95%CI, 32–49%), found in 14 articles performed in Asia. However, there were no significant differences of positive culture rate between continents (*p* = 0.211).

### 2.3. Distribution of Bacteria Isolated from Corneal Lesions

Within 14,596 samples reported from 38 studies, 15,350 bacterial strains isolated from corneas of BK cases were summarized (Table 2). Gram-positive cocci were the major type of bacteria (62%, 58–67%), followed by Gram-negative bacilli (30%, 26–33%), Gram-positive bacilli (5%, 4–7%), and Gram-negative cocci (5%, 4–7%). The five most common bacterial organism detected was *Staphylococcus* spp. (41.4%, 36.2–46.7%), *Pseudomonas* spp. (17.0%, 13.9–20.7%), *Streptococcus* spp. (13.1%, 10.9–15.7%), *Corynebacterium* spp. (6.6%, 5.3–8.3%) and *Moraxella* spp. (4.1%, 3.1–5.4%). Figure 4 presented the increasing trends of Gram-positive cocci and the decreasing trends of Gram-negative bacilli in 1996–2015. In 2000s, the proportion of Gram-positive cocci exceeded that of Gram-negative bacilli. The upward trend of Gram-positive cocci and the downward trend of Gram-negative bacilli were both significant (z = 1.71, *p* = 0.04; z = −1.88, *p* = 0.03). At the genus level, the percentage of *Pseudomonas* spp. declined from 39.9% (1990s) to 12.2% (2000s) and *Staphylococcus* spp. rose from 25.9% (1990s) to 39.5% (2000s) (Figure 4). The upward trend of *Staphylococcus* spp. was significant (z = 1.71, *p* = 0.04) and the downward trend of *Pseudomonas* spp. was not significant (z = −1.22, *p* = 0.22).

### 2.4. Antibiotic Susceptibility of the Bacterial Strains Isolated from Corneal Lesions

All results of drug susceptibility tests of the strains were summarized in Figure 5. Most Gram-positive cocci were susceptible to aminoglycoside (86%, 3916 sensitive in 4527), following cephalosporin and fluoroquinolone, 79% (1997 sensitive in 2515) and 81% (3921 sensitive in 4831), separately. A resistance of macrolides was observed (57%, 212 sensitive in 375). As for Gram-negative bacilli, most isolates were susceptible to cephalosporin and fluoroquinolone (cephalosporin: 96%, 1269 sensitive in 1328; fluoroquinolone: 96%, 2519 sensitive in 2611), followed by aminoglycoside (92%, 2547 sensitive in 2783). Figure 6 shown the changing trends of susceptibility of pathogens to common drugs. In Gram-positive cocci, the susceptibility to common antibiotics such as cefazolin, gatifloxacin, moxifloxacin and ofloxacin showed a decreasing trend since 2010. For Gram-negative bacilli, a susceptibility over 90% was maintained in all recommended antibiotics [24] in recent years.

## 3. Discussion

Bacterial keratitis is the second most common cause of legal blindness worldwide [25]. Although broad-spectrum antibiotics, such as levofloxacin, have always been used to control BK, more targeted treatment was required to improve the clinical outcomes [26]. However, the spectrum of pathogenic bacteria and their susceptibility to antibiotics varied with the different regions. Kaye et al. showed that these variations were related to the latitude and the degree of urbanization of the population studied [26]. Therefore, local epidemiology of bacterial spectrum and its resistance to antibiotics should be paid attention and is mandatory to know. In this study, 30-year changing trends of microbiological profile of BK were reviewed.

From our results, the average rate of positive culture from the samples of BK was 47%. The large difference in positive culture rates (21~83%) between literatures may be related to the different indications of corneal scrapes in different medical institutions, the ability of microbiology laboratory, or to the fact that the study population had already received topical antibiotics treatment before corneal scraping.

In this study, the common bacterial isolates were *Staphylococcus* spp., *Pseudomonas* spp., *Streptococcus* spp., *Corynebacterium* spp. and *Moraxella* spp., which was consistent with studies in the USA [27], UK and Canada. Although the number of *Moraxella* strains was 311, lower than that in *Serratia* strains (373), *Moraxella* spp. were isolated from 7121 samples (4.4%, 16 studies) and 373 *Serratia* spp. were isolated from 10,431 samples (3.6%, 27 studies). Thus, we concluded that *Moraxella* spp. were more common than *Serratia* spp. *Pseudomonas* spp. was demonstrated to be the most common pathogen in Singapore and Malaysia, which may result from a local high frequency of using contact lenses. Keya et al. found the percentage of *Enterobacterales* was 15.3%, higher than the percentage of *Staphylococcus aureus* and *Pseudomonas aeruginosa*. In addition, the upward trend of Gram-positive cocci and the downward trend of Gram-negative bacilli were observed, especially for the upward trend of *Staphylococcus* spp. and the downward trend of *Pseudomonas* spp. at the genus level. Similar trends were also presented in the studies from the UK [6] and Iran [23]. CoNS, one of the most common strains of *Staphylococcus* spp., was the major bacteria of normal skin, including eye lid. It would have more opportunity to contaminate the cornea; in addition, the conduction of corneal scraping was also easily susceptible to its contamination. The decreasing percentage of *Pseudomonas* spp. may be attributed to the wide application of some antibiotics, such as tobramycin and fluoroquinolones, and to the improvement in health conditions. Though several studies [4,5,6] reported a significant increase of *Moraxella* keratitis in UK for the last two decades, the trend was not shown in this worldwide study. Perhaps this trend was limited to specific regions.

In Gram-positive cocci, most isolates were susceptible to aminoglycoside (86%, 3916 sensitive in 4527), and resistant to macrolides in more than half of drug susceptivity tests (57%, 212 sensitive in 375). In the other two classes of antibiotics commonly used in clinical practice against Gram-positive cocci, cephalosporin and fluoroquinolone, 79% (1997 sensitive in 2515) and 81% (3921 sensitive in 4831) isolates showed susceptivity. Since 2010, the susceptibility of Gram-positive cocci to common antibiotics such as cefazolin, gatifloxacin, moxifloxacin and ofloxacin has shown a decreasing trend. For Gram-negative bacilli, most isolates were susceptive to cephalosporin (96%, 1269/1328) and fluoroquinolones (96%, 2519/2611), followed by aminoglycoside (92%, 2547/2783). The trends of susceptibility seemed stable and maintained above 90%. Some studies had reported a group of *Pseudomonas* spp. with multiple drug resistance [28,29]. It is necessary to use targeted antibiotics in case of the development of resistant strains.

The meta-analysis revealed the trend and distribution of bacterial keratitis pathogens and their drug resistance pattern guiding ophthalmologists to diagnosis and to choosing antibiotics based on their local data. Our study also possessed some limitations. Original studies often reported their results via time period (e.g., 2000–2005 CoNS 30%), not specific year (e.g., 2000 CoNS 30%), which affected the accuracy of our study. Differences between original studies, such as culture methods, participants who received topical antibiotics treatment before corneal scraping and experience of clinics would increase the difficulty of pathogen distribution analysis. Our study could still provide useful guidance for clinics.

In conclusion, the worldwide average positive culture was 47% between 2000 to 2020. The percentage of Gram-positive cocci was increasing, and the percentage of Gram-negative bacilli was decreasing. The five most common bacterial organism were *Staphylococcus* spp., *Pseudomonas* spp., *Streptococcus* spp., *Corynebacterium* spp. and *Moraxella* spp. Increasing trends of Gram-positive cocci and the decreasing trends of Gram-negative bacilli were observed in 1996–2015. Most Gram-positive cocci were susceptive to aminoglycoside and were resistant to macrolides. Gram-negative bacilli were sensitive to cephalosporin, fluoroquinolone and aminoglycoside and maintained susceptibility above 90%. The decreasing trends of susceptibility were observed in Gram-positive cocci to most common antibiotics. Ophthalmologists should pay attention to the changing trends of pathogen distribution and their drug resistance patterns and modify the diagnosis and choose sensitive antibiotics based on the local data.

## 4. Materials and Methods

### 4.1. Databases and Search Strategy

Four databases, including Embase, Medline, Web of Science, and CINAHL, were searched, and publication date of articles were limited between January 2000 to December 2020. Main key words were “bacterial keratitis”, “culture results” and “drug resistance”. The whole search strategy was (“bacterial keratitis” OR “infectious keratitis” OR “microbial keratitis” OR “bacterial infections” OR “corneal ulcers” OR “bacterial infections of cornea”) AND (“organisms” OR “culture results” OR “isolates” OR “microbiology” OR “antibiotic susceptibility” OR “resistance pattern”). In addition, the document type was restricted to “article”, the language was restricted to “English”, and the subjects were restricted to “human”. More details of the search strategy could be found in Appendix A.

### 4.2. Literature Selection and Quality Assessment

The literature searched in the above databases was imported into the EndNote X 9 software library for merging and de-duplicating; then, the titles and abstracts were screened by two researchers (Z.Z., J.L.) according to the following inclusion criteria: (1) Purpose of the article should concentrate on reporting the distribution and resistance pattern of bacterial keratitis isolates; (2) Subjects of the literature must contain patients suspected of infective keratitis and confirmed by positive bacterial culture results; (3) Pathogens were isolating from corneal scrapes via culture and were identified at least at the genus level; (4) Drug susceptibility tests should be conducted for isolated strains via in vitro minimum inhibitory concentration testing (MIC) or the disk diffusion method. The homogeneity of our meta-analysis was controlled by exclusive criteria below: (1) Subjects with small sample size or specific risk factors; (2) Subjects already selected by authors would be excluded for positive rate analysis. (3) Multiple articles published using the same data would be deduplicated. After preliminary exclusion of unrelated references, we downloaded the full text of each citation; the literature quality was evaluated based on a methodological scoring system of “rate” [30]. The detailed quality criteria were as follows: (1) Whether there is a clear diagnostic basis for bacterial keratitis; (2) Whether the sample size (the number of bacterial culture specimens) is greater than 246 cases; (3) Whether there are clear criteria for positive bacterial culture; (4) Whether there are clear study parameters, such as positive rate or drug susceptibility results; (5) Whether the data is complete. Complete data should include the description of study populations, methods for the drug susceptibility test, and protocol for bacteria separating and identifying. Each criterion was given one score, and studies with a score of 4 or more were of high quality and included for analysis. All the steps of screening and quality assessment were carried out independently by two researchers (Z.W., X.X.). In case of disagreement over the inclusion of the literature, a third, more experienced researcher (Q.L.) would make the final decision.

### 4.3. Data Extraction

According to the purpose of this study, the data extraction scale of the literature was developed. For each included article, four aspects of information would be extracted. The first is the basic information of the literature and the institution of the author, including the publication year, time that the research was conducted, author, title, medical institution, etc. Next, the necessary data of positive rate of culture, bacterial strains distribution and drug susceptibility were extracted. The positive rate of culture was defined as the proportion of BK patients with positive culture results in all culture-treated patients suspected of infective keratitis. The parameter necessary for analyzing bacterial strain distribution contained the total number of strains isolated, number classified by Gram staining and number of strains as accurate as possible to species. Parameters related to drug susceptibility included the number of susceptible or resistant species performed on a certain species of bacterium to a certain drug. All above data were extracted into Microsoft Excel software.

### 4.4. Statistical Analysis

Data were analyzed with SPSS software (SPSS for windows, version 16.0, SPSS, Chicago, IL, USA). The meta-analysis was conducted using R program (V.4.0, R Foundation for Statistical Computing, Vienna, Austria) with meta package. All effect sizes were transformed into a single common metric, event rate with its 95% confidence interval, which indicated the number of participants in each sample endorsing bacterial keratitis. Either a random effects model or a fixed effects model was used to perform meta-analysis, which was determined by I^2^ statistic; I^2^ > 50% indicates a large heterogeneity among included studies and, correspondingly, a random effects model would be used; otherwise, a fixed effects model would be used.

## Figures and Tables

**Figure 1 antibiotics-11-00238-f001:**
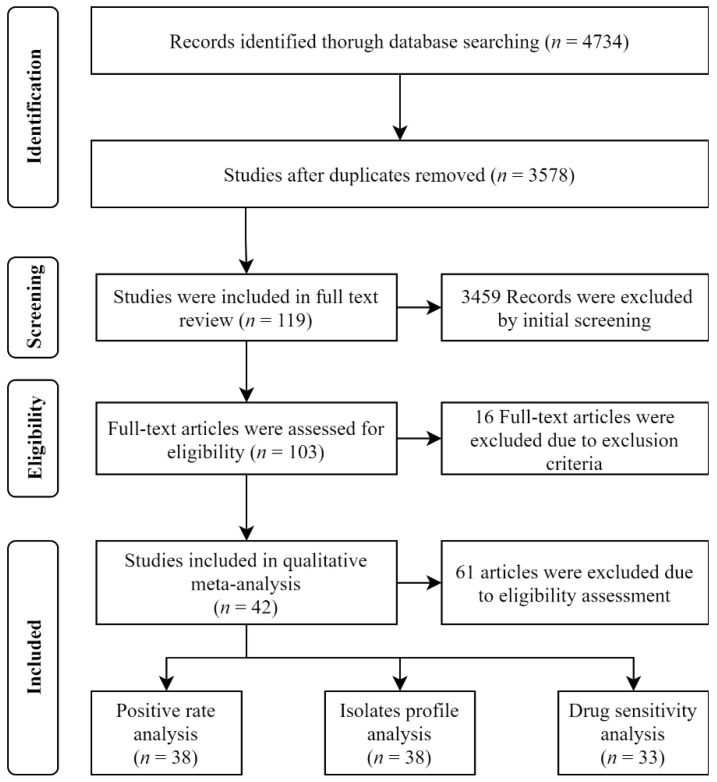
The PRISMA flow chart of paper selection.

**Figure 2 antibiotics-11-00238-f002:**
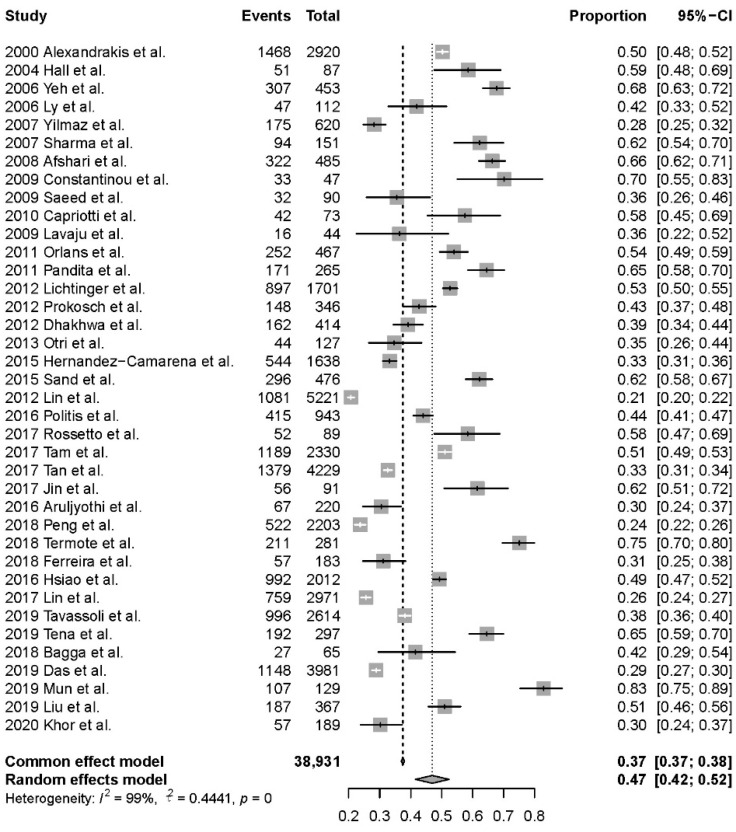
Analysis of positive rate of bacterial culture (38 studies).

**Figure 3 antibiotics-11-00238-f003:**
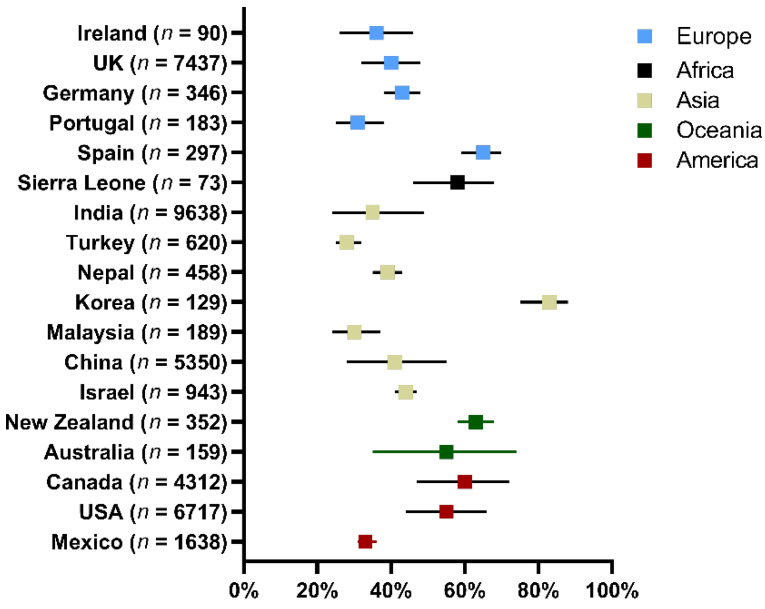
Positive rate of bacterial culture from corneal lesions in different regions.

**Figure 4 antibiotics-11-00238-f004:**
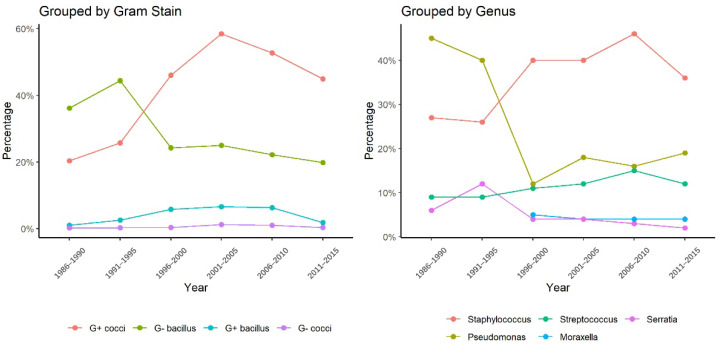
The changing trends of bacterial isolates from corneal lesions in 1990s–2020s.

**Figure 5 antibiotics-11-00238-f005:**
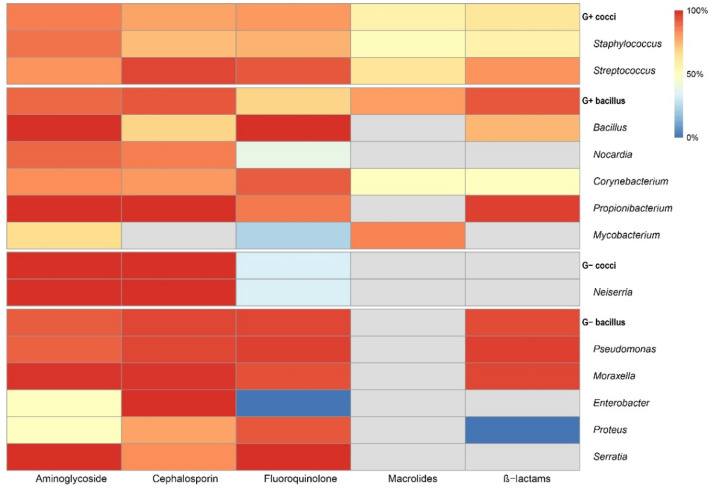
Results of drug susceptibility test of the strains isolated from corneal lesions.

**Figure 6 antibiotics-11-00238-f006:**
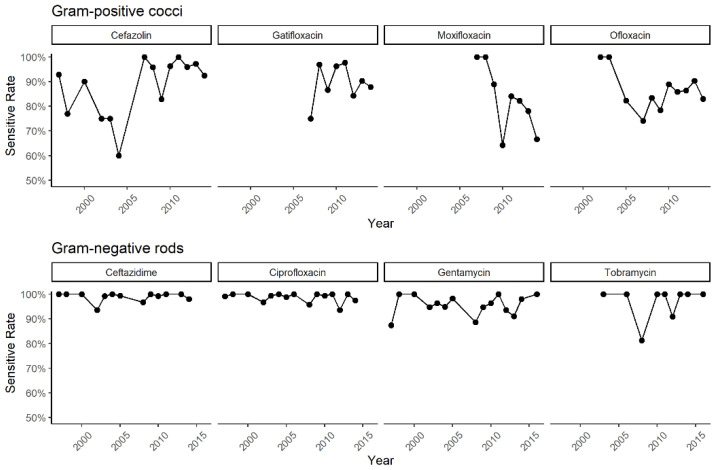
The changing trends of drug susceptibility in 1990s–2020s.

**Table 1 antibiotics-11-00238-t001:** Characteristics of the studies included in the meta-analysis.

Authors (Years)	Country	City	Study Period	Sample Size	Positive Rate (%)	Microbiological Profiles
**Europe**						
Schaefer (2001)	Switzerland	Lausanne	1997–1998	85	86	*Staphylococcus epidermidis* (29.0%) *Staphylococcus aureus* (16.0%) *Pseudomonas* species (7.0%)
Saeed (2009)	Ireland	Dublin	2001–2003	90	36	*Pseudomonas* species (33.3%) Coagulase negative *staphylococci* (12.1%) *Staphylococcus aureus* (9.0%)
Orlans (2011)	UK	Oxford	1999–2009	467	54	Coagulase negative *staphylococci* (25.8%) *Pseudomonas aeruginosa* (24.3%) *Staphylococcus aureus* (14.3%)
Prokosch (2012)	Germany	Münster	2002–2009	346	43	*Staphylococcus aureus* (31.7%) *Pseudomonas* species (7.5%) *Streptococcus pneumoniae* (6.0%)
Otri (2013)	UK	Nottingham	2007–2007	129	35	*Staphylococcus aureus* (18.8%) *Pseudomonas aeruginosa* (15.0%) *Pneumococcus* (9.4%)
Tan (2017)	UK	Manchester	2004–2015	4229	30	Coagulase negative *staphylococci* (38.5%) *Pseudomonas* (37.1%) *Staphylococcus aureus* (23.9%)
Ferreira (2018)	Portugal	Porto	2007–2015	235	38	*Staphylococcus aureus* (23.1%) *Corynebacterium macginleyi* (20.0%) *Pseudomonas aeruginosa* (13.8%)
Tavassoli (2019)	UK	Bristol	2006–2017	2116	38	Coagulase negative *staphylococci* (49.9%) *Pseudomonas* species (22.0%) *Streptococci* (9.7%)
Tena (2019)	Spain	Guadalajara	2010–2016	298	65	Coagulase *negative staphylococci* (28.6%) *Cutibacterium* species (19.6%) *Corynebacterium* species (9.8%)
**Africa**						
Capriotti (2010)	Sierra Leone	Freetown	2005–2006	73	58	*Pseudomonas aeruginosa* (39.7%) *Staphylococcus aureus* (27.4%) Coagulase negative *staphylococci* (5.5%)
**Asia**						
Sharma (2007)	India	Hyderabad	2002–2002	170	62	*Staphylococcus epidermidis* (18.6%) *Streptococcus pneumoniae* (18.6%) *Pseudomonas* species (4.9%)
Yilmaz (2007)	Turkey	Izmir	1990–2005	620	28	*Staphylococcus epidermidis* (26.6%) *Staphylococcus aureus* (24.4%) *Streptococcus pneumoniae* (15.5%)
Fong (2007)	China	Taipei	1994–2005	272	-	*Pseudomonas aeruginosa* (46.7%) *Cutibacterium* species (8.1%) *Nontuberculous Mycobacteria* (6.6%)
Lavaju (2009)	Nepal	Dharan	2007–2008	44	36	*Staphylococcus aureus* (70.0%) *Pseudomonas* species (15.0%) *Acinetobactor* species (5.0%)
Feilmeier (2010)	Nepal	Kathmandu	2006–2009	468	15	*Streptococcus pneumoniae* (69.0%) *Staphylococcus aureus* (11.0%) *Staphylococcus epidermidis* (7.0%)
Dhakhwa (2012)	Nepal	Siddharthanagar	2007	414	39	*Staphylococcus epidermidis* (29.6%) *Streptococcus viridans* (15.1%) *Pseudomonas aeruginosa* (14.0%)
Lin (2012)	India	Madurai	2006–2009	5221	21	*Staphylococcus epidermidis* (31.9%) *Pseudomonas aeruginosa* (12.4%) *Staphylococcus simulans* (5.5%)
Politis (2016)	Israel	Jerusalem	2002–2014	943	44	Coagulase-negative *staphylococci* (43.9%) *Pseudomonas aeruginosa* (24.8%) *Streptococcus pneumoniae* (6.9%)
Hsiao (2016)	China	Taoyuan	2003–2012	2012	40	*Pseudomonas aeruginosa* (24.4%) Coagulase-negative *staphylococci* (16.6%) *Cutibacterium* species (9.1%)
Aruljyothi (2016)	India	Madurai	2011–2013	234	30	*Pseudomonas aeruginosa* (37.9%) *Streptococcus pneumoniae* (24.1%) *Staphylococcus aureus* (12.0%)
Lin (2017)	China	Guangzhou	2009–2013	2973	12	*Staphylococcus epidermidis* (31.9%) *Pseudomonas aeruginosa* (12.4%) *Staphylococcus simulans* (5.5%)
Bagga (2018)	India	Hyderabad	1991–2012	60	42	*Staphylocci* (35.0%) *Corynebacteria* (25.5%) *Streptococci* (24.0%)
Mun (2019)	Korea	Seoul	2007–2016	129	78	Coagulase negative *staphylococci* (15.9%) *Staphylococcus aureus* (12.1%) *Pseudomonas aeruginosa* (10.3%)
Liu (2019)	China	Taipei	2007–2016	363	51	*Pseudomonas* species (44.7%) *Nontuberculous Mycobacteria* (7.5%) *Propioebacterium* species (6.8%)
Das (2019)	India	Hyderabad	2007–2014	3981	29	*Streptococcus pneumoniae* (16.1%) *Staphylococcus aureus* (13.8%) *Pseudomonas* species (7.4%)
Khor (2020)	Malaysia	Sarawak	2010–2016	221	30	*Pseudomonas aeruginosa* (33.6%) *Staphylococcus aureus* (3.4%) *Corynebacterium* species (1.7%)
**Oceania**						
Hall (2004)	New Zealand	Christchurch	1997–2001	87	59	Coagulase negative *staphylococci* (19.3%) *Moraxella* species (19.3%) *Coryebacterium* species (16.0%)
Ly (2006)	Australia	Sydney	2002–2003	112	42	Coagulase negative *staphylococci* (38.0%) *Pseudomonas aeruginosa* (21.0%) *Corynebacterium* species and coryneform bacteria (15.0%)
Constantinou (2009)	Australia	Melbourne	1998–2007	47	70	*Pseudomonas aeruginosa* (33.3%) Coagulase negative *staphylococci* (11.1%) *Cutibacterium acnes* (8.9%)
Pandita (2011)	New Zealand	Hamilton	2007	265	65	Coagulase negative *staphylococci* (40.8%) *Staphylococcus aureus* (11.5%) *Streptococcus pneumonia* (7.5%)
Watson (2019)	Australia	Sydney	2016	224	75	Coagulase negative *staphylococci* (47.8%) *Staphylococcus aureus* (9.6%) *Pseudomonas aeruginosa* (9.6%)
**America**						
Alexandrakis (2000)	USA	Miami	1990–1998	2920	50	*Pseudomonas aeruginosa* (25.7%) *Staphylococcus aureus* (19.4%) *Serratia marcescens* (7.6%)
Yeh (2006)	USA	Durham	1997–2004	453	68	Coagulase *negative staphylococci* (39.0%) *Staphylococcus aureus* (12.0%) *Pseudomonas* species (10.0%)
Afshari (2008)	USA	Boston	1999–2000	485	66	Coagulase *negative staphylococci* (45.5%) *Staphylococcus aureus* (15.2%) *Diphtheroids* (5.7%)
Lichtinger (2012)	Canada	Toronto	2000–2010	1701	53	Coagulase *negative staphylococci* (37.0%) *Staphylococcus aureus* (17.0%) *Streptococcus* species (17.0%)
Hernandez-Camarena (2015)	Mexico	Mexico City	2002–2011	1638	33	*Staphylococcus epidermidis* (25%) *Pseudomonas aeruginosa* (12%) Coagulase negative *staphylococci* (10%)
Sand (2015)	USA	Los angeles	2008–2012	476	62	Coagulase *negative staphylococci* (51.4%) *Pseudomonas aeruginosa* (15.3%) *Staphylococcus aureus* (12.8%)
Rossetto (2017)	USA	Miami	1992–2015	107	58	*Pseudomonas aeruginosa* (42.1%) *Strenotrophomonas maltophilia* (17.5%) *Serratia marcescens* (8.8%)
Tam (2017)	Canada	Toronto	2000–2015	2330	49	Coagulase negative *staphylococci* (37%) *Staphylococcus aureus* (15%) *Streptococcus species* (15%)
Jin (2017)	USA	Houston	2011–2015	96	62	*Pseudomonas aeruginosa* (33.9%) Coagulase negative *staphylococci* (26.8%) *Streptococcus pneumoniae* (10.7%)
Peng (2018)	USA	San Francisco	1996–2015	2203	24	*Staphylococcus aureus* (25.1%) Coagulase negative *staphylococci* (20.5%) *Streptococcus viridans* (13%)
Termote (2018)	Canada	Vancouver	2006–2011	281	75	Coagulase negative *staphylococci* (25.6%) *Streptococcus* species (12.4%) *Staphylococcus aureus* (12.1%)

**Table 2 antibiotics-11-00238-t002:** Distribution of bacteria isolated from corneal lesions of bacterial keratitis.

Organism	Isolates	Percentage (%)	95%CI (%)
**Gram-positive cocci**	**8786**	**62.3**	**57.9~66.5**
*Staphylococcus*	5311	41.4	36.2~46.7
*Streptococcus*	1913	13.1	10.9~15.7
*Gemella*	18	3.8	2.4~6.0
*Micrococcus*	41	2.5	1.8~3.3
*Kocuria*	12	1.6	0.9~2.8
*Enterococcus*	10	1.3	0.7~2.4
*Aerococcus*	7	0.8	0.3~1.6
*Leuconostoc*	6	0.8	0.4~1.7
*Peptostreptococcus*	2	0.7	0.2~2.9
**Gram-negative bacilli**	**3776**	**29.6**	**26.0~33.5**
*Pseudomonas*	2331	17.0	13.9~20.7
*Moraxella*	311	4.1	3.1~5.4
*Serratia*	373	3.4	2.7~4.2
*Haemophilus*	64	2.2	1.8~2.8
*Proteus*	54	2.1	1.1~4.0
*Escherichia*	38	2.0	1.5~2.7
*Klebsiella*	17	1.8	1.1~2.8
*Achromobacter*	1	1.9	0.0~12.2
*Acinetobacter*	26	1.8	1.3~2.7
*Burkholderia*	18	1.8	1.2~2.9
*Enterobacter*	13	1.2	0.7~2.0
*Stenotrophomonas*	11	1.1	0.6~2.0
*Citrobacter*	2	1.0	0.3~3.9
*Morganella*	5	0.9	0.4~2.0
**Gram-positive bacilli**	871	5.2	3.9~6.8
*Corynebacterium*	284	6.6	5.2~8.3
*Nocardia*	96	3.9	2.4~6.0
*Cutibacterium*	243	3.3	1.7~6.0
*Bacilli*	184	2.6	0.7~8.5
*Sphingomonas*	2	2.6	0.7~8.5
*Brevibacterium*	2	2.6	0.7~8.5
*Clostridium*	2	1.0	0.3~3.9
*Mycobacterium*	10	0.8	0.4~1.5
*Aeromonas*	3	0.8	0.3~2.1
**Gram-negative cocci**	26	5.2	3.9~6.8
*Neisseria*	5	0.8	0.3~1.9
**not mentioned**	1891	11.9	9.3~15.1

## Data Availability

Not applicable.

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
