# Peer review of "Pathogens and Antibiotic Susceptibilities of Global Bacterial Keratitis: A Meta-Analysis"

_antibiotics, 2022, doi:10.3390/antibiotics11020238_

Round 1

Reviewer 1 Report

Peer review of “The global isolated pathogens and antibiotic susceptibilities of bacterial keratitis: a meta-analysis,” by Zhang et al, for consideration by the MDPI journal Antibiotics.

This manuscript describes a met-analysis of the bacterial keratitis and reports the type of bacteria that cause infections, the positive culture rate, antibiotic susceptibility, and trends over time. Included studies span the globe and increase the value of this study.

Overall this is a very well written manuscript with clear inclusion parameters; however, the discussion would benefit from careful editing (specific criticisms and suggestions are given below). The authors effectively argue that the results can be used to help guide antibiotic therapy choices in places that do not work up the infecting microbes during keratitis, which is unfortunately prevalent. The overall study may fit better in a an ophthalmology journal, but it also fits with Antibiotics.

Comments in order of appearance in the study:
1. Please replace “sensitivity” with “susceptibility” throughout, I believe this is more appropriate for characterizing bacterial response to antibiotics

2. Table 1, consider adding the city because in a large country like China or Australia, the location can highly influence climate and the spectrum of bacteria that cause infection.

3.Please help the reader to differentiate what is different between Results section 2.2 and 2.3 (Figures 2 and 3). Please clarify.

4. Table 2. The number of Strains for Moraxella 311 and for Serratia is 373, yet Moraxella is listed as more common than Serratia - please clarify. It might be worth noting this in the discussion if the n is higher for Serratia.

5. Table 2. Consider “Isolates” rather than “strains” for Table 2.

6. Update the bacterial names to current taxonomic standards - for example the Enterobacteria aerogenes are now classified as Klebsiella aerogenes; Propionibacterium acnes has been renamed as Cutibacterium acnes; line 141 - many of the group of bacteria listed here as Enterobacteriaceae have been renamed as the Enterobacterales (Proteus, Serratia).

7. Figure 6 is not mentioned in the text. Please remove the figure or add the relevant information describing the analysis.

8. Important, but common mix-up is to describe antibiotics as sensitive/susceptible rather than the bacteria. For example, line 155, “the most sensitive type of drug was aminoglycoside” - the drug was not tested for sensitive, the bacteria were sensitive to the drug.

9. Line 179-180 - see point 8. The drug is not sensitive. Please rewrite. 

10. Discussion section, minor grammatical errors
    Line 148. Add “the” before cornea
    Line 161. Gram-negative bacteria “were” rather than “was”
    Line 169. Suggest using “specific year” rather than “time point”
    Line 173. Move “still” between “could” and “provide”.
    Line 188. “bacilli” is the plural - please check through out.

11. Line 26. Please define what “data is complete” means. For example, one could write: “whether the data was complete with respect to x, y, and z” - fill in x, y, and z.

Author Response

1. Please replace “sensitivity” with “susceptibility” throughout, I believe this is more appropriate for characterizing bacterial response to antibiotics

Author response: Thank you for your suggestions. We have replaced all “sensitivity” with “susceptibility”.

2. Table 1, consider adding the city because in a large country like China or Australia, the location can highly influence climate and the spectrum of bacteria that cause infection.

Author response: I agree with your suggestions. The cities of large countries were added and could be detected in table 1.

3. Please help the reader to differentiate what is different between Results section 2.2 and 2.3 (Figures 2 and 3). Please clarify.

Author response: Thank you for your suggestions. In Result 2.2, “Positive rate of culture” described the ratio of culture positive samples to all suspected IK samples, while in Result 2.3 “Distribution of bacteria isolated from corneal lesions” presented bacterial spectrum in those positive samples for bacteria culture. Please find these explanations in Line 74-76 “A total of 38,931 samples scraping from the cornea of BK patients were reviewed in the study. Among them, 14,596 samples were culture positive and positive rate of cul-ture was 47% (95%CI, 42%-52%) based on the 38 studies (Figure 2)” and Line 90-91 “Within 14,596 samples reported from 38 studies, 15,350 bacterial strains isolated from corneas of BK cases were summarized (Table 2)”.

4. Table 2. The number of Strains for Moraxella 311 and for Serratia is 373, yet Moraxella is listed as more common than Serratia - please clarify. It might be worth noting this in the discussion if the n is higher for Serratia.

Author response: Thank you for your comments. We clarified this point in Line 142-145 in discussion section. “Although the number of Moraxella strains was 311, lower than the number of Serratia strains (373), Moraxella spp. were isolated from 7,121 samples (4.4%, 16 studies) and 373 Serratia spp. were isolated from 10,431 samples (3.6%, 27 studies). So, we concluded Moraxella spp. were more common than Serratia spp. in the isolated strains from BK lesions”

5. Table 2. Consider “Isolates” rather than “strains” for Table 2.

Author response: Thank you for your suggestion. We have already replaced the word “strains” with “isolates” in table 2.

6. Update the bacterial names to current taxonomic standards - for example the Enterobacteria aerogenes are now classified as Klebsiella aerogenes; Propionibacterium acnes has been renamed as Cutibacterium acnes; line 141 - many of the group of bacteria listed here as Enterobacteriaceae have been renamed as the Enterobacterales (Proteus, Serratia).

Author response: Thank you for your suggestions. We have already updated bacterial names from “Propionibacterium” to “Cutibacterium” and from “Enterobacteriaceae” to “Enterobacterales”.

7. Figure 6 is not mentioned in the text. Please remove the figure or add the relevant information describing the analysis.

Author response: Thank you for your comments. We have already cited “Figure 6” in Result 2.4 section (See Line 115).

8. Important, but common mix-up is to describe antibiotics as sensitive/susceptible rather than the bacteria. For example, line 155, “the most sensitive type of drug was aminoglycoside” - the drug was not tested for sensitive, the bacteria were sensitive to the drug.

Author response: Thank you for your suggestions. We have already modified this part according to your comments. See Line 109-120. “Most Gram-positive cocci were susceptible to aminoglycoside (86%, 3916 sensitive in 4527), following the cephalosporin and fluoroquinolone, 79% (1997 sensitive in 2515) and 81% (3921 sensitive in 4831) separately. A resistance of macrolides was observed (57%, 212 sensitive in 375). As for Gram-negative bacilli, most isolates were susceptible to cephalosporin and fluoroquinolone … For Gram-negative bacilli, a susceptibility over 90% was maintained in all recommended antibiotics[27] in recent years.

9. Line 179-180 - see point 8. The drug is not sensitive. Please rewrite.

Author response: Thank you for your suggestions. We have already modified this part according to your comments. See Line 161-168. “In Gram-positive cocci most isolates were susceptible to aminoglycoside (86%, 3916 sensitive in 4527), and resistant to macrolides in more than half of drug susceptivity tests (57%, 212 sensitive in 375). In the other two classes of antibiotics commonly used in clinical practice against Gram-positive cocci, cephalosporin and fluoroquinolone, 79% (1997 sensitive in 2515) and 81% (3921 sensitive in 4831) isolates shown susceptivity … For Gram-negative bacilli, most isolates were susceptive to cephalosporin.

10. Discussion section, minor grammatical errors

    Line 148. Add “the” before cornea

    Line 161. Gram-negative bacteria “were” rather than “was”

    Line 169. Suggest using “specific year” rather than “time point”

    Line 173. Move “still” between “could” and “provide”.

Line 188. “bacilli” is the plural - please check through out.

Author response: Thank you for your comments. We have corrected these grammatical errors in Line 154, 161, 176, 180, and replaced all “bacillus” with “bacilli”.

11. Line 26. Please define what “data is complete” means. For example, one could write: “whether the data was complete with respect to x, y, and z” - fill in x, y, and z.

Author response: Thank you for your suggestions. In method section (Line 223-225), “data is complete” was defined as follows: “A complete data should include the description of study populations, methods for the drug susceptibility test, and protocol for bacteria separating and identifying.

Reviewer 2 Report

Well done job.

Author Response

Thank you for your recognition.

Reviewer 3 Report

In this study was conducted a meta-analysis to review the global culture rate, distribution, current trends, and drug sensitivity of isolates from BK over the past 20 years. The aim was to value the spectrum of pathogenic bacteria and their sensitivity to antibiotics varied with the different regions. The results were more interesting. This article is very interesting for the public of this journal. The language of the paper includes many errors.

1) In the Abstract, you have not presented materials and methods and the city and time of this study. Improve it.

2) …gram-negative bacilli (30%), gram-positive bacilli (5%)… Gram should be capitalized, Gram-negative, correct it il all text.

3) Line 36: “pneumoniae, et al[15-19]” ?? correct it.

4) Line 41: https://doi.org/10.1155/2020/8847812, add this study as a reference and in the main text.

Author Response

1. In the Abstract, you have not presented materials and methods and the city and time of this study. Improve it.

Author response: Thank you for your comments. In the part of abstract, we presented materials and methods, the city and time of our study (Line 11-14). “Four databases were searched, and published date was limited between 2000 and 2020. Main key words were “bacterial keratitis”, “culture results” and “drug resistance”. 42 studies from 21 countries (35 cities) were included for meta-analysis.

2. …gram-negative bacilli (30%), gram-positive bacilli (5%)… Gram should be capitalized, Gram-negative, correct it il all text.

Author response: Thank you for your suggestions. We have capitalized the word “Gram-positive” and “Gram-negative” in all text.

3. Line 36: “pneumoniae, et al[15-19]” ?? correct it.

Author response: Thank you for your suggestion. We have modified this in Line 39. “Streptococcus pneumoniae, and others [17-21]”

4. Line 41: https://doi.org/10.1155/2020/8847812, add this study as a reference and in the main text.

Author response: Thank you for your suggestion. We have added this study (https://doi.org/10.1155/2020/8847812) as reference 2 in the main text (Line 28-29). “Infectious keratitis (IK) is a potentially sight-threatening condition, which leads to at least 1.5 to 2 million new cases of unilateral blindness every year [1-3].”

  1. Petrillo, F., V. Folliero, B. Santella, et al., Prevalence and Antibiotic Resistance Patterns of Ocular Bacterial Strains Isolated from Pediatric Patients in University Hospital of Campania “Luigi Vanvitelli,” Naples, Italy. International Journal of Microbiology 2020. 2020, 8847812.

Reviewer 4 Report

This study is very interesting because the meta-analysis conducted is very large and a to review the global culture rate, distribution, current trends, and drug sensitivity of isolates from bacterial keratitis over the past 20 years. The results were more interesting, but the paper includes many errors.

- The title needs to be changed! the global isolated ?? correct it.

- Line 33: “to make empirical diagnosis to know the pathogenic 33 spectrum of BK and provide appropriate therapy to eradicate microorganism”, improve and corretc this concept, …. to make empirical diagnosis to know the pathogenic microorganism and antibiotics appropriate for eradicating the infection…

- Line 39: “pathogen of BK was. Pseudomonas spp. Was”, correct it

- PMID: 34063711, add this study as a reference.

- Line 36: “Coagulase-negative staphylococci (CNS)”, for me is better reported CoNS such acronym, substitute it in all text.

Author Response

1. The title needs to be changed! the global isolated ?? correct it.

Author response: Thank you for your suggestion. The title of this article was changed into “Pathogens and antibiotic susceptibilities of global bacterial keratitis: a meta-analysis”.

2. Line 33: “to make empirical diagnosis to know the pathogenic 33 spectrum of BK and provide appropriate therapy to eradicate microorganism”, improve and corretc this concept, …. to make empirical diagnosis to know the pathogenic microorganism and antibiotics appropriate for eradicating the infection…

Author response: Thank you for your suggestion. This sentence had been modified in Line 36-37. “Therefore, it is important for clinicians to make empirical diagnosis to know the pathogenic microorganism and antibiotics spectrum of BK and provide appropriate therapy to control infection [15].”

3. Line 39: “pathogen of BK was. Pseudomonas spp. Was”, correct it

Author response: Thank you for your suggestion. This sentence had been modified in Line 39-42. “The bacterial spectrum and drug susceptibility for bacterial isolates from different areas during different periods are widely reported [5, 8, 18, 22, 23], but the most common pathogen of BK remains debatable.”

4. PMID: 34063711, add this study as a reference.

Author response: Thank you for your suggestion. We have added this study (PMID: 34063711) as reference 3 in the main text (Line 28-29). “Infectious keratitis (IK) is a potentially sight-threatening condition, which leads to at least 1.5 to 2 million new cases of unilateral blindness every year [1-3].”

  1. Petrillo, F., D. Pignataro, F.M. Di Lella, et al., Antimicrobial Susceptibility Patterns and Resistance Trends of Staphylococcus aureus and Coagulase-Negative Staphylococci Strains Isolated from Ocular Infections. Antibiotics (Basel) 2021. 10, 527.

5. Line 36: “Coagulase-negative staphylococci (CNS)”, for me is better reported CoNS such acronym, substitute it in all text.

Author response: Thank you for your suggestion. The acronym “CNS” had been changed into “CoNS” in all text.

Round 2

Reviewer 1 Report

The authors' have addressed my comments and the revision is improved. This is a nice addition to the field.